# Do bank voles (*Myodes glareolus*) trapped in live and lethal traps show differences in tick burden?

**Nicolas De Pelsmaeker**[1]*, **Lars Korslund**[2], **Øyvind Steifetten**[1]

1 Department of Nature, Health and Environment, University of Southeastern Norway, Bø, Norway,
2 Department of Natural Sciences, University of Agder, Kristiansand, Norway

* nicolas.de.pelsmaeker@usn.no

**Data Availability Statement:** All data files are available from the USN figshare database (url: https://figshare.com/s/1893989238abdd5b6acd, doi:10.23642/usn.12416012).

## Abstract

In studies assessing tick abundance, the use of live traps to capture and euthanize rodent hosts is a commonly used method to determine their burden. However, captive animals can experience debilitating or fatal capture stress as a result prior to collection. An alternative method is the use of lethal traps, but this can potentially lead to tick drop-off between the time of capture and collection. In this study, in order to determine whether subjecting animals to capture stress is inevitable, we tested the difference in sheep tick (*Ixodes ricinus*) larval burdens between bank voles (*Myodes glareolus*) captured alive and euthanized, and lethally trapped bank voles. During 2017 and 2018, 1318 bank voles were captured using live (Ugglan Special no. 2) and lethal (Rapp2 Mousetrap) traps during two consecutive years over three seasons in two locations in Norway. Voles captured alive would remain captive until euthanized, while lethally trapped voles were killed instantly upon capture. Log-linear models, accounting for overdispersion, were used to determine whether trap type was influencing observed tick burden. Bank voles captured in lethal traps carried 5.7% more larvae compared to euthanized voles captured in live traps, but this difference was not significant (p = 0.420). Males were overall captured 2.7 times more frequently than females, and the sex ratio was equal in both trap types. This study shows that the use of lethal traps to determine tick burden of rodents is sufficiently reliable, without having to subject animals to potentially lethal stress, hereby reducing some ethical concerns of animal suffering and the results thereof, without compromising accuracy. Lethal trapping is also often more economical and practical, further favoring this collection method.

## Introduction

In the pursuit of reliable data in animal studies, killing animals is sometimes necessary and unavoidable, and without such practices many aspects of animal ecology, demography, physiology and biology would remain unstudied [1]. It allows for a detailed study of the captured animal and can reveal much more about the biology of a specimen than from a live examination, and it also permits the collection of animals for later dissection and preservation, and

**Funding:** The author(s) received no specific funding for this work.

**Competing interests:** The authors have declared that no competing interests exist.

offers possibilities that are not available if an animal is to remain intact or alive. However, wild animals that are first captured and later euthanized may still experience some degree of stress from the time spent in captivity and from handling by the researcher [2]. Trap induced stress responses usually occur within minutes of capture [3], and longer periods of captivity lead to increasingly more stress [4, 5]. Animals have shown to develop capture myopathy [6–8] which often has fatal consequences [9], and animals that are kept in captivity after capture can die from secondary stress responses [10]. Consequently, there are ethical considerations in the practice of capture, and efforts should be made to reduce the discomfort of captive animals.

In parasitological studies, the parasite burden of small mammals can be used as a proxy for parasite distribution and infestation rates, as small mammals act as hosts for a number of parasitic species, including ticks [11–13]. One of the main hosts of larval ticks are small rodents, which are considered good indicators of tick abundance as they occupy the same physical space [14]. Small rodents can act as reservoirs for zoonotic pathogens [15–18], and generalist ectoparasites such as ticks can further transmit these pathogens to humans [19–21]. Some of these pathogens can be dangerous to humans [22, 23]. Ticks can themselves be hosts to a number of pathogens and readily transmit these pathogens to new hosts [24–26], and thus rodent parasites are an important element in the study of epidemiology [27]. One method often used when studying the distribution and prevalence of ticks is to measure the burden on live hosts in the field before returning them [11–13], but this is very stressful for the animal being handled, and it can be difficult to accurately determine the precise numbers in this way due to sub-optimal conditions, leading to imprecise measurements. Most such studies involve visual inspection and mechanical removal of the parasites from the animal using a comb or a tweezer, while physically restraining it or using anesthetics [21, 28–31]. Estimating and removing ticks from a live host is particularly complicated as ticks are cemented into the skin surface while feeding [32]. The unavoidable and intensive handling of an animal during examination can also be a major cause of stress in itself [33, 34], and may potentially lead to the death of the animal [9]. A more reliable alternative to live examination is to euthanize the captured animals when collected, in order to examine them thoroughly in a controlled environment (hereafter referred to as euthanized captures). Although this method allows for a more reliable assessment of ticks on hosts [35], this does not prevent any stress related discomfort experienced by the captive animal between time of capture and collection. An option to reduce capture-induced stress in small mammals such as rodents is to employ lethal traps, where the animal is killed instantly upon capture (hereafter referred to as lethal capture). When studying ticks however, this might constitute a problem, as ticks require a live host in order to complete a blood meal. It is possible that upon the death of a host, ticks will start to progressively abandon the dead host (drop-off) in order to molt or to quest for a new host in an attempt to complete the blood meal (refeeding) [36, 37]. During field studies, there can be a considerable amount of time (several hours) between a lethal capture and collection of the animal. Therefore, it is possible that during that time period at least some of the ticks will drop off after the host dies, and the number of ticks found on the animal may not represent the total burden at the time of death. It is unknown which cues (or lack thereof) might cause ticks to detach from a dead host, but in laboratory conditions ticks have been observed to start detaching from mice three hours after death [36].

In this study, we investigated if larval tick burden size on small rodents differed between euthanized captures and lethal captures to determine whether live trapping followed by euthanasia is necessary to reliably assess the total tick burden on small rodents, or whether the use of lethal traps is sufficient for this purpose. The aim was to determine if lethal trapping could be a viable alternative to euthanized captures in order to reduce the stress animals are likely to experience in live traps. Because little is known on how ticks behave after the host dies, we had

no clear expectation on the outcome of the study, and we could not predict which trapping method would yield the largest tick burdens.

## Materials and methods

### Study area

The study was part of a larger research project investigating the distribution of ticks along an altitudinal gradient, conducted in two separate areas in Norway during 2017 and 2018. The first study area was a southern facing mountain slope on the Lifjell massif (N59˚ 26.495' E9˚ 02.603') near Bø i Telemark. Lifjell is located within the boreonemoral to southern boreal zone, characterized by mixed coniferous and deciduous forest, and is characterized by a continental climate. The second study area was the northern facing Erdal valley (N61˚ 05.817' E7˚ 24.688') near Lærdalsøyri (hereafter referred to as Lærdal) close to the innermost part of the Sognefjorden fjord. It is located in the middle to northern boreal zone, and lies approximately 150 km east of the western coastline. The vegetation is dominated by deciduous forest and it has a coastal climate.

### Host trapping

This study was carried out in strict accordance with the regulations issued by the Norwegian Environment Agency, and a permit was provided prior to the start of the sampling (Miljødirektoratet, reference number: 2017/4651) for the duration of the trapping period. The protocol for capturing animals was approved by the Animal Ethics Committee of the Department of Nature, Health and Environment (University of South-Eastern Norway). All efforts were made to minimize animal suffering.

At each study area, 10 trapping stations were established along an altitudinal gradient from 100 up to 1000 m. a. s. l., at every 100 m altitude interval. At every altitude station, two capture plots were constructed. One with 20 live traps for euthanized captures (Ugglan Special Nr. 2, Grahnab AB, Sweden; www.grahnab.se) (Fig 1A) and the other with 20 lethal traps (Rapp2

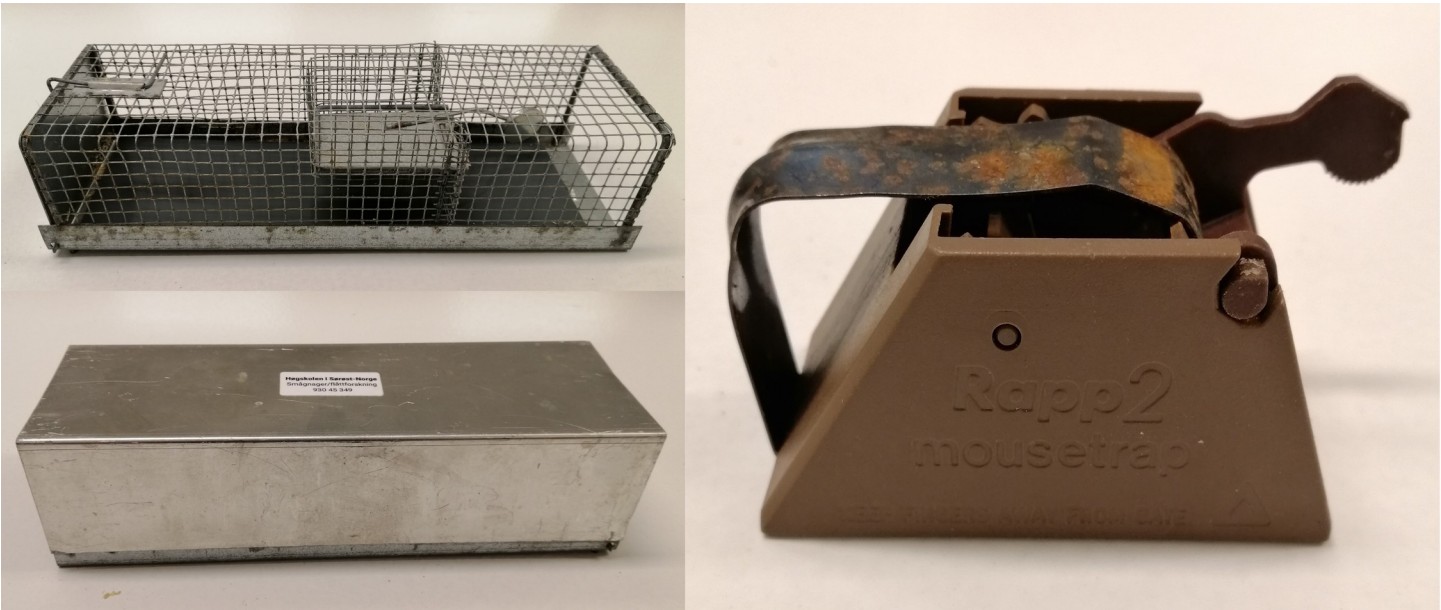

**Fig 1. The two trap types used in the study.** (A) Ugglan special Nr 2 live trap (top), covered with an Ugglan special long roof metal cover (bottom). (B) Rapp2 lethal mousetrap.

Mousetrap, www.rappfellene.no) (Fig 1B). Rapp2 mousetraps were chosen over classic snap traps as they reduce the risk of non-lethal capture (e.g. leg or tail), which can cause considerable suffering. Lethal trapping killed the animals instantly by cervical fracturing. Both trap types were arranged in a 4 by 5 grid, with 10 meters spacing between each trap. Live and lethal plots were spaced a minimum of 100 m apart to avoid home range overlap of hosts [38, 39]. Live traps were baited with a slice of apple for hydration and whole oats for caloric value, and a lining of sawdust was provided as insulation on the trap floor. Lethal traps were baited with peanut butter for practical reasons as it is easily applied to the inside of the trap body. The trapping of small mammals took place three times per year, during spring (May 20th– 30th), summer (July 20th– 30th), and autumn (September 20th– 30th). The only exception to this schedule was during the spring season of 2017, when capturing took place from June 1st until June 7th in both study areas, and only up to 700 m. a. s. l., as there was too much snow in both areas to permit trapping earlier and above this altitude. Traps were checked for captures once every 24 hours during the trapping period, and collection of the captures started at 8h30 every day. Triggered traps were rebaited and reset. Handling rodents and examining them for ticks in the field can be stressful and can cause harm or injury to the animals [40], and it is difficult to accurately determine tick burden on live small mammals [41]. Therefore, we opted for a postmortem full body examination, and voles captured in live traps were euthanized and collected. Full body examinations of both euthanized and lethal captures provided a higher degree of sensitivity [42]. Individuals captured in live traps were euthanized in the field by cervical dislocation of the head, and sealed in separately coded plastic bags. Cervical dislocation of the head was also instantly fatal. After every collection day, all animals were placed in a freezer at –20˚C. Because bank voles (*Myodes glareolus*) constituted more than 86% of all rodents captured, we only used bank voles in the analysis.

Humidity and temperature have a direct influence on tick activity, and are important drivers of phenological patterns and host-seeking behavior [43]. For this reason, a temperature logger (TinyTag Plus 2 –TGP 4017) was placed inside a DataMate instrument cover (ACS-5050), and mounted on a pole 50 cm above ground level in between live and lethal plots at every altitude station. These loggers recorded temperature at a 1-hour interval for the duration of the study period (June 2017 –September 2018), as a measure of environmental conditions throughout the transect.

## Laboratory processing

After every field season the captured bank voles were checked for ticks. The voles were taken out of the freezer the evening before examination, and left to thaw in a cold room at 10˚C overnight. The animals were then removed from the plastic bags and the empty bags were checked for ticks that might have dropped off. Wet animals were first dried using a hairdryer to more easily detect the ticks. The voles were then processed one by one, and ticks were removed, counted and placed in a 1.5mL plastic vial containing a 70% ethanol solution (1 vial per host). Ticks found attached or just present on the host were removed using a tweezer. The hosts were then checked starting with the head, ears and snout, followed by neck and throat, back and abdomen, legs, feet and tail. Finally a lice comb was used over the whole body of the animal from tail to head (against the hair orientation), and the animal was subsequently shaken by the tail during 5 seconds above a white plastic tray to collect any ticks that were potentially missed during the examination. The minimum time needed to process one vole was 20 minutes. The hosts were also weighed to the nearest 10th of a gram and their sex was determined. Once the examination was completed the processed animals were bagged in new plastic bags and refrozen at -20˚C.

Individual ticks were determined for life stage and species using the determination keys published by Arthur [44] and Hillyard [45] under a Zeiss Discovery V20 stereomicroscope. Because more than 80% of all ticks collected were sheep ticks (*Ixodes ricinus*), and of these more than 94% were larvae, only *I. ricinus* larvae were used for this study. A minority of vole ticks (*I. trianguliceps*) was also encountered, but was not used in this analysis, as it is a rodent specialist solely living in burrows of the host [46], and is not encountered in open vegetation. Ticks that were too damaged to allow for precise identification were discarded from the study altogether. After determination, the ticks were replaced in vials for long-term storage.

## Data analysis

All statistical analyses were performed using R version 3.5.3 [47]. We used general linear regression models, using larval burden (i.e. the number of tick larvae on an individual) as the response variable. To compare the larval burden between live and lethal traps, trap type was used as a categorical predictor. Additionally, study area (Lærdal and Lifjell), collection year (2017 and 2018), season (spring, summer and autumn) and daily average temperature (˚C) were used as extrinsic covariates, and the individual characteristics host weight (in grams) and sex (male or female) were used as intrinsic covariates. To investigate if ambient temperature could affect any drop off of ticks from the captured host, temperature was averaged for a 24-hour period spanning from 12:00 AM the day before capture until 12:00 AM on the day of capture. Two-way interactions between all covariates and trap type were considered, and we used a stepwise backwards model selection approach. We started with a full model containing all predictors and interactions, and progressively removed interactions or predictor variables that were not significant in a type II ANOVA test, until a nested model only yielded significant predictor variables. Larval burden on bank voles was a count, and we therefore used a Poisson distribution in the glm-function to model the data. Overdispersion was detected, hence we corrected the standard errors using a quasi–GLM model according to Zuur et al. [48], where the variance is given by $\varphi \times \mu$, where $\mu$ is the mean and $\varphi$ the dispersion parameter [48]. To visually represent the relationships between larval burden and the predictor variables, we used effect plots from the *ggplot2* package [49]. To test whether there was a trap selection for a particular trap type, we regressed host trappability by using trap type as a binomial response variable and the host specific variable sex and weight as predictors, adding an overdispersion parameter $\varphi$ [48]. A Wilcoxon rank sum test was performed to test if the body mass of the captured voles differed between sexes. A p-value $< 0.05$ was considered significant.

## Results

For both years and study areas a total of 43920 trap nights were performed, capturing 1318 bank voles (Table 1). Other captured rodents were field voles (*Microtus agrestis*), tundra voles (*M. oeconomus*), grey red-backed voles (*Myodes rufocanus*), wood mice (*Apodemus sylvaticus*), yellow-necked mice (*A. flavicollis*) and house mice (*Mus musculus*). Three species of shrews

**Table 1. Number of bank voles captured per trap type in each study area during 2017 and 2018.**

| Site | 2017 | | | | | | 2018 | | | | | | |
|---|---|---|---|---|---|---|---|---|---|---|---|---|---|
| | Spring | | Summer | | Autumn | | Spring | | Summer | | Autumn | | |
| | Live | Lethal | Live | Lethal | Live | Lethal | Live | Lethal | Live | Lethal | Live | Lethal | Total |
| **Lifjell** | 61 | 22 | 61 | 108 | 107 | 54 | 11 | 3 | 6 | 2 | 3 | 6 | 444 |
| **Lærdal** | 55 | 48 | 117 | 109 | 64 | 163 | 31 | 37 | 57 | 70 | 29 | 94 | 874 |
| **Total** | 116 | 70 | 178 | 217 | 171 | 217 | 42 | 40 | 63 | 72 | 32 | 100 | 1318 |

**Table 2. Number of larval *Ixodes ricinus* ticks collected from bank voles per trap type in each study area during 2017 and 2018.** Below are the mean (SD) burdens per vole.

| Site | 2017 | | | | | | 2018 | | | | | |
|---|---|---|---|---|---|---|---|---|---|---|---|---|
| | Spring | | Summer | | Autumn | | Spring | | Summer | | Autumn | |
| | Live | Lethal | Live | Lethal | Live | Lethal | Live | Lethal | Live | Lethal | Live | Lethal |
| Lifjell | 144 | 63 | 36 | 182 | 18 | 20 | 21 | 3 | 0 | 0 | 21 | 3 |
| | 2.4 ± 6.5 | 2.9 ± 6.0 | 0.6 ± 1.6 | 1.7 ± 3.8 | 0.2 ± 0.7 | 0.4 ± 1.1 | 1.9 ± 5.7 | 1.0 ± 1.7 | 0.0 ± 0.0 | 0.0 ± 0.0 | 7.0 ± 6.1 | 0.5 ± 0.8 |
| Lærdal | 678 | 772 | 527 | 527 | 32 | 328 | 389 | 340 | 197 | 270 | 18 | 146 |
| | 12.3 ± 11.5 | 16.1 ± 21.3 | 4.5 ± 6.5 | 4.8 ± 11.0 | 0.5 ± 1.4 | 2.0 ± 5.1 | 12.5 ± 18.7 | 9.2 ± 15.6 | 3.4 ± 8.2 | 3.9 ± 9.3 | 0.6 ± 1.3 | 1.6 ± 2.2 |

were also captured: common shrews (*Sorex araneus*), pigmy shrews (*S. minutus*) and water shrews (*Neomys fodiens*). In both study areas fewer captures were made in 2018 compared to 2017 (413 and 31 in Lifjell, 556 and 318 in Lærdal for 2017 and 2018, respectively). In total 4735 *I. ricinus* larvae were collected, and 47.0% of the captured voles were infested with at least one larvae. Infestation rates between capture methods were similar (44.4% and 49.3% of euthanized and lethally trapped voles, respectively). Larval burden ranged from 1 to 100, and voles were infested with on average 3.6 larvae. Mean burdens per study area and year are summarized in Table 2.

Voles that were lethally trapped had 5.7% larger tick burdens in comparison to euthanized captures, but the effect of trap type alone was not significant (Table 3), and trap type was only retained in the model due to the significant interaction between trap type and season (F = 4.13, df = 2, p = 0.02). Larval burdens on bank voles trapped in lethal traps were somewhat higher in spring and autumn, and burdens were nearly equal between traps in summer (Fig 2). Burdens differed however substantially and significantly between seasons overall (F = 186.89, df = 2, p< 0.001), showing the largest burdens in spring, with a nearly linear decline throughout the trapping period for both euthanized and lethal captures (Fig 2). A post-hoc Wilcoxon rank sum test showed no significant differences in tick burdens between capture types in each season, with the exception of autumn (W = 26371, p< 0.001, n = 520). The final model, best describing the variation in larval burden between individual bank voles, included an additive effect of trap type, study area, year, season, host sex and temperature, as well as an interaction between trap type and season (Table 3). Study area was the most influential predictor of larval burden, as it differed significantly between study areas (F = 158.10, df = 1, p< .001), and was

**Table 3. Estimated regression parameters, standard errors, t-values and p-values for the final model, describing the factors influencing larval burdens on bank voles.**

| | Estimate | Std. Error | *t*-value | p-value |
|---|---|---|---|---|
| Intercept | 0.234 | 0.304 | 0.772 | 0.440 |
| Site Lærdal | 1.395 | 0.176 | 7.915 | < 0.001 |
| Year 2018 | -0.585 | 0.143 | -4.081 | < 0.001 |
| Season Summer | -1.477 | 0.190 | -7.780 | < 0.001 |
| Season Autumn | -1.567 | 0.214 | -7.329 | < 0.001 |
| Temperature | 0.101 | 0.023 | 4.436 | < 0.001 |
| Sex Female | -0.281 | 0.125 | -2.243 | 0.025 |
| Trap type Live: Season Autumn | -0.864 | 0.444 | -1.944 | 0.052 |
| Trap type Live | -0.121 | 0.150 | -0.806 | 0.420 |
| Trap type Live: Season Summer | -0.092 | 0.232 | -0.396 | 0.692 |

Non-significant predictors are retained in the model because of their involvement in an interaction or other predictor variable which is significant.

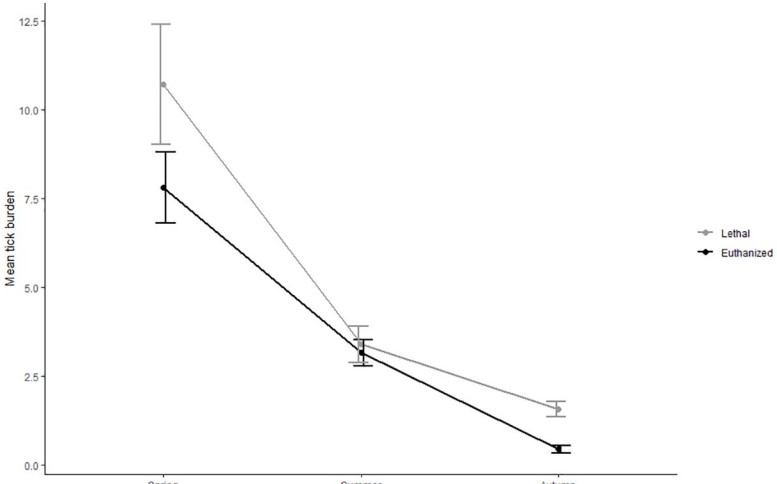

**Fig 2. Mean tick burden of *I.ricinus* on lethally captured and euthanized bank voles per season.** Mean tick burdens are for both collection years, study areas and all altitudes combined. Error bars represent standard errors.

overall more than four times higher in Lærdal compared to Lifjell (4.8 and 1.2 ticks per vole, respectively). The only exception was in autumn 2018, where mean burden in Lifjell was twice that of Lærdal [2.6 ± 4.1(sd) and 1.3 ± 2.1(sd), respectively]. Overall, burdens were significantly lower in 2018 compared (F = 33.51, df = 1, p< 0.001).

Temperature had a significant positive effect on burden (F = 36.82, df = 1, p< 0.001), and this effect was equally strong irrespective of trap type, as shown by the non-significant interaction between temperature and trap type (F = 0.06, df = 1, p = 0.802). Males carried overall more ticks than females (F = 9.93, df = 1, p = 0.002). The Wilcoxon rank sum test showed that the difference in body mass between the trap types was statistically significant (W = 109980, p < 0.001, n = 1318). Voles captured in lethal traps were significantly heavier than voles captured in live traps (F = 7.86, df = 1, p = 0.005), but the mean difference was only 0.9 g. Male voles were captured 2.7 times more frequently than females (Fig 4), and the body mass of captured females (Mdn = 25.9) was higher than that of males (Mdn = 20.3). The sex of the captured vole was not significantly different in either trap type (F = 0.14, df = 1, p = 0.707).

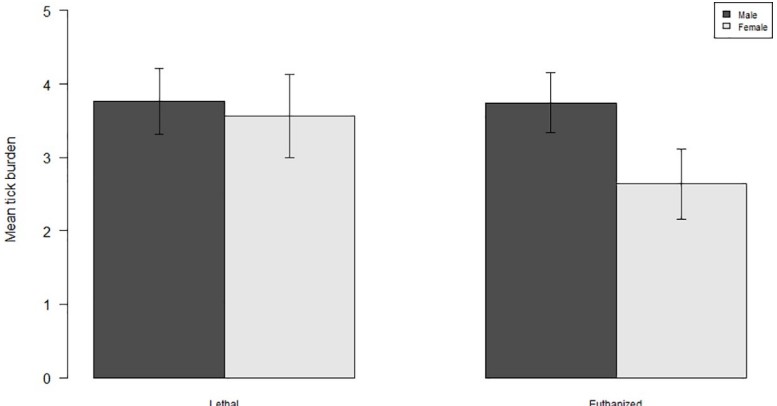

**Fig 3. Mean tick burden of *I. ricinus* on bank voles per trap type and vole sex for lethal and euthanized captures.** Mean tick burdens are for both collection years, study areas and all altitudes combined. Error bars represent standard errors.

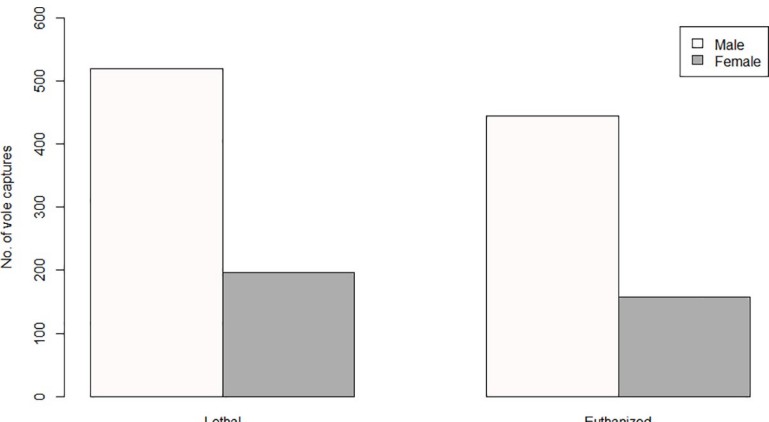

**Fig 4. Total number of male and female voles captured per trap type.** Total captures are for both collection years, study areas and all altitudes combined.

## Discussion

This study aimed to determine if larval tick burdens differed between euthanized voles captured in live traps and voles killed in lethal traps. Our data showed that there was no significant difference in the number of ticks between bank voles that were lethally trapped, and those that remained alive until collection, suggesting that the use of lethal traps does not result in a different drop-off rate compared to live traps. In fact, our data showed that, if any, lethally trapped voles had borderline significantly higher burdens during autumn in comparison to euthanized voles.

Potential drop-off rates from dead hosts can be a deciding factor in selecting which trapping method to employ, and research on this process is limited. Nakao and Sato [36] found that Taiga ticks (*I. persulcatus*) began to abandon dead laboratory mice three hours after time of death, and Piesman [37] demonstrated that deer tick larvae (*I. dammini*) abandoned hamsters during two days following the death of the host. However, these studies represented laboratory conditions and it is reasonable to assume that the field conditions in our study (i.e. diurnal and nocturnal temperature fluctuations, as well as differences in humidity and light regime) could have affected the behavior of larvae and the duration of attachment after the death of a host. It may have taken a certain amount of time for larvae to decide to leave a dead host as the body of hosts probably cooled down at different rates based on environmental conditions. Attached larvae might have continued to feed for a little while even after death. Larvae might also have detached, but remained in the fur of a dead host for a time before dropping off completely. Investigating detachment rates under field conditions could prove valuable in understanding the cues ticks use to determine host death and their decision to abandon and/or attempt refeeding. Several studies have documented tick drop-off rates of live hosts when ticks reach full engorgement under laboratory conditions [31, 50, 51], and it is possible that under field conditions the time voles spent captive in live traps provided a window of opportunity for a fraction of the attached larvae to have reached full engorgement and dropped off the host before it was collected, and hence would not have been counted during the laboratory assessment. Both live and lethal traps would have allowed for engorged larvae to leave. Dizij and Kurtenbach [52] found that the longest attachment duration for *I. ricinus* larvae on bank voles was 79 hours. Once an animal entered a live trap, it was isolated from the environment and probably did not acquire new ticks. With a maximum of 24 hours between live capture and euthanasia, this means that potentially over a third of the burden could potentially have

reached full engorgement and have dropped off within that period (24 hours = 30,4% of 79 hours). More realistically, bank voles are most active right before dusk, and right after dawn [53], so the time spent in captivity was more likely to be either 9–12 hours, meaning a potential drop off of 11–15% in the time captive before euthanasia, or about 2–5 hours, resulting in a drop off of 2–6% of the burden at the time of collection. The mean burden in live traps was 7% lower to lethal traps and 12% fewer larvae were collected in total. However, voles captive in live traps may have had time to remove some of the ticks through grooming, potentially explaining the lower larval burdens found in live traps.

Females carried on average fewer larvae than males (Fig 3), hence larvae dropping off females during captivity could have made a larger proportional difference compared to males. Pollock [55]] found that western black-legged ticks (*I. pacificus*) fed faster and dropped off quicker on female lizards (*Sceloporus occidentalis*) during the mid-summer season [54]. During spring, male voles carried on average twice as many larvae compared to females (10.5 and 4.9 respectively). This could indicate that not only would the proportional difference in drop-off be larger in females compared to males, but that the difference drop-off in females could be exacerbated by faster feeding and less time required for ticks to reach full engorgement. Multiple studies have found that tick infestation rates are skewed towards male rodent hosts [14, 55–59]. Bank vole males have larger home ranges than females [60, 61], and therefore roam around more than females. Hence, the likelihood of males to encounter questing ticks is higher than for females. In particular, *I. ricinus* larvae do not disperse far from the place where the adult female tick oviposited the egg batch, and hence their distribution is spatially clumped. Because of differences in roaming behavior, males have a higher probability of encountering aggregated tick larvae, resulting in larger burdens. In fact, the five highest burdens encountered in this study (n = 81, 81, 84, 97, 100) all occurred on males. In general, the sex ratio of bank voles is close to 1:1 [62], or fluctuating between sexes with generational turnover [63, 64]. The difference in activity between the sexes can help explain why males were more trappable than females, and the ratio between male and female captures was nearly identical in both trap types, indicating that trap type itself did not have an effect on the likelihood of a vole of either sex to enter the trap. Although captured females were heavier than males, and voles in lethal traps were heavier than voles in live traps, we do not consider that trap selection was biased between the sexes, given the fact that the weight difference between trap types was small. Live traps baited with apple slices and oats seems to perform equally well compared to lethal traps baited with peanut butter.

The main factor driving larval burden on bank voles in this study was location, indicating that environmental factors such as temperature and humidity, and overall host community, are more influential on burden than trap type. Ticks are sensitive to environmental factors such as temperature and humidity, and due to the relative surface to volume ratio, larvae are expected to be particularly sensitive to desiccation [65]. Both study areas differ in climate, and possibly the local climatic conditions are contributing to the overall abundance of ticks. Adults of *I. ricinus* quest for hosts higher in the vegetation, compared to subadult life stages [66], and tend to prefer larger mammalian hosts such as roe deer (*Capreolus capreolus*) and red deer (*Cervus elaphus*) [25, 67]. Performing a survey of cervids was outside the scope of this study, and it is possible that differences in local abundances of larger hosts may contribute to a higher abundance of larvae. Roe deer is more abundant in eastern Norway (Lifjell), whereas red deer is more abundant in western Norway (Lærdal) (Reimers et al., 1990 and Solberg et al., 2009, as cited in Handeland et al. [68]). A local survey of cervids or other large mammalian hosts may help elucidate whether adult tick host availability is influential on the larval infestation rates of voles.

Burden increased with temperature, and this is in accordance with other studies, in which above a certain tolerance limit in humidity, temperature can increase the activity of questing ticks [68–70], and can subsequently result in an increase in larval burden. The effect of temperature on tick burden was however, almost identical in both trap types, showing little influence of the trap type itself in relation to temperature. Although temperature had a positive effect on tick burdens, 2018 was not only warmer, but also drier than 2017 in both study areas (84 and 37 days of precipitation less in Lifjell and Lærdal, respetively; data: Norwegian Meteorological Institute), and this decrease in humidity may explain why overall burdens were lower in 2018 compared to 2017. The seasonal effects of tick abundance and rates of tick parasitism are well documented, and are dependent on life stage and locality. Questing nymphs and adults generally show a bimodal activity pattern during spring and autumn, and less activity during the summer [21, 71], whereas larvae generally show a single activity peak in summer to autumn [71–77]. Burdens on hosts tend to follow the same pattern for each life stage in rodents [55, 78]. Larval burdens in this study indeed showed a single spring peak in May, and a decline through summer (Fig 4). Norway being relatively cool and humid during the summer, ticks may not have been limited by dry conditions and high temperatures. The seasonal patterns found in this study might be due to ticks that have not found a host in spring, continuing questing during the summer until low autumn temperatures limited their activity.

## Conclusions

In the past decades, animal welfare has become increasingly important in animal studies [1, 2, 79–81], and efforts are continuously being made to reduce the discomfort or suffering animals might experience during research activities. Although killing the animals is sometimes necessary to achieve research goals, efforts should nonetheless be made to prevent or minimize avoidable suffering. Here we show that lethal traps are just as reliable as live traps in assessing tick burdens on rodents, and by avoiding the stress experienced by animals in live traps before euthanasia, lethal traps are the preferred choice. Given the fact that in this study, live captures were euthanized when collected, and both capture methods resulted in the death of the animal, we argue that in the scope of animal welfare, the use of lethal traps is the preferred option. An argument could be made that lethal traps might behave differently than live traps regarding trappability and random captures, thereby making them less suitable for ecological studies. However, our results show that both trap types are equally effective at capturing bank voles, and that intrinsic factors such as the weight or sex of the animal does not influence the likelihood of a vole to enter either trap type. Lethal traps are generally also cheaper than live traps, and usually smaller, lighter, less bulky, and quicker to operate, which further adds to the benefits of using lethal traps. In conclusion, we propose lethal traps should be considered as an alternative to live capture and euthanasia to prevent unnecessary suffering of the animals, without sacrificing reliability in tick burden assessment on rodents.

## Acknowledgments

Part of the field work in Lærdal was performed on private land. We would like to thank Britt Bjørkum, who kindly granted us access to her property for trapping, and who was very helpful in communicating our efforts to other landowners in the valley, and helping us with on-site logistics.

## Author Contributions

**Conceptualization:** Lars Korslund, Øyvind Steifetten.

**Formal analysis:** Nicolas De Pelsmaeker, Lars Korslund.

**Supervision:** Lars Korslund, Øyvind Steifetten.

**Writing – original draft:** Nicolas De Pelsmaeker.

**Writing – review & editing:** Lars Korslund, Øyvind Steifetten.

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
