## [Decision Letter · Decision Letter 0]

30 Jun 2020

PONE-D-20-16817

Do bank voles (Myodes glareolus) trapped in live and lethal traps show differences in tick burden size?

PLOS ONE

Dear Dr. De Pelsmaeker,

Thank you for submitting your manuscript to PLOS ONE. After careful consideration, we feel that it has merit but does not fully meet PLOS ONE’s publication criteria as it currently stands. Therefore, we invite you to submit a revised version of the manuscript that addresses all of the points raised during the review process.

We look forward to receiving your revised manuscript.

Kind regards,

Brian Stevenson, Ph.D.

Academic Editor

PLOS ONE

Journal Requirements:

"This study was funded by the University of South-Eastern Norway."

3. Please ensure that you refer to Figure 3 in your text as, if accepted, production will need this reference to link the reader to the figure.

Reviewers' comments:

Reviewer's Responses to Questions

**Comments to the Author**

1. Is the manuscript technically sound, and do the data support the conclusions?

Reviewer #1: Yes

Reviewer #2: Yes

Reviewer #3: Yes

2. Has the statistical analysis been performed appropriately and rigorously? 

Reviewer #1: Yes

Reviewer #2: Yes

Reviewer #3: Yes

3. Have the authors made all data underlying the findings in their manuscript fully available?

Reviewer #1: No

Reviewer #2: Yes

Reviewer #3: Yes

4. Is the manuscript presented in an intelligible fashion and written in standard English?

Reviewer #1: Yes

Reviewer #2: Yes

Reviewer #3: Yes

5. Review Comments to the Author

Reviewer #1: An article considering the impact of trapping methodology on the ability to collect ticks from Voles. Also discuss predictive factors on tick abundance.

I have the following concerns that I would like to see addressed:

- Convince us about Why you are interested in ticks on voles in the first place. What pathogens would they be suggested to play a role in transmitting? If something such as Borrelia burgdorferi., then larvae are not likely to be infected due to lack of transovarial transmission. Discuss. What evidence is there of voles being a reservoir for tick-borne pathogens?

- Why were voles euthanized following capture and tick assessment? Explain why ticks couldn't be collected and then the animal released as a viable option. Improve the introductory rationale.

- How was stress in the live-captured animals measured? How do you know they were experiencing stress being held for a period of time.

- What were the species other than bank voles encountered during trapping?

- What other species of tick (other than Ixodes ricinus) were detected on the rodents?

- If as found, only larvae would be expected on voles, explain when peaks of Ixodes ricinus larvae generally occur.

- To avoid differences in behaviour or tick prevalence of populations in the areas sampled, I would have used a mixed grid design, rather than putting lethal and live traps in separate areas.

- Weblink to data sharing wasn't functioning; but maybe this will be activated upon publication.

Minor corrections are requested, as suggested below:

TITLE: The word 'size' isn't needed, and to state burden is sufficient. This applies throughout the manuscript.

Short Title: Suggest 'Comparative tick burdens on voles by trap type'

I would even consider renaming the main title as "Influences on Ixodes ricinus larval tick burden on bank voles (Myodes glareolus) captured via lethal and live traps".

ABSTRACT: check where commas are needed;

- Check the spelling of 'ricinus';

- (line 14) - add the word 'tick' before burden.

Since you only analyzed larval ticks, please state that within the abstract/title.

METHODS: Line 123 - Replace 'for this study' with 'in the analysis'.

RESULTS: Your key hypothesis/question concerns the trap type effect, so I would move those results up; then later talk about your models of tick abundance factors.

Line 221: remove the 'd' (before Fig 4).

TABLE 3: Include in the table legend what the model is of. Also, it may be my print out but, there seems an overlap of text in the final column

DISCUSSION:

Reads a little sporadic overall, like a long chain of thought, but mentions some good points.

- Line 254-55: I'd remove this sentence.

- Page 11: Did your live trap prevent replete ticks from leaving the trap area? For example, Fig.1 indicates that the covered Ugglan / Sherman-type trap was used. Was the inside of traps examined for ticks which had dropped off? / Could the closed trap retain ticks (and humidity) better than the lethal trap.

- Lines 284-287: This sentence seems to discuss two separate topics, and could be separated.

- Mention grooming behavior e.g. of live captured mice, or grooming differences between sexes.

- Did you consider controlling for humidity in your traps (to avoid drop-off)?

- How would tick life stage be expected to affect if there was a difference between lethal and live trap tick burdens across the seasons? You mention adults on deer, but what about nymphs?

- Lines 312-313: Suggest 'Performing a survey of cervids was outside the scope of this study,'

- Please discuss what your finding of a significant burden difference in study area implies? e.g. should different regions of the country be examined.

- Lines 319-320: I'm wondering how you can state this, when your methodology only describes one temperature recorder placed between the two grid types.

- Lines 326-329: Check this long sentence. Line 329 - replace 'continued' with 'continuing'.

- In general, check places where commas should be inserted.

- Line 334: Still haven't read why killing the animals is necessary for this particular tick monitoring activity.

FIGURES:

I realize myself what the axes/legends are referring to, but suggest including making this obvious. For example, label '"tick" burden', rather than burden; label 'No. of "Vole" Captures'. Also describe more fully within legends what is being shown in the figure/table.

Reviewer #2: Authors aimed to test whether trapping method, lethal or live traps, used to capture bank voles (Myodes glareolus) had significant impacts on the tick burden size at time of collection. The study was conducted as a sub-study of a larger project over a period of 2 years in Norway and decidedly focused only on infestation burden of Ixodes ricinus larvae on the captured rodents. They found that while lethal traps did result in larger burden sizes, a significant difference was not obtained (p = 0.420) and suggest the use of lethal over live traps to avoid causing the animal stress during capture.

While the study did mention that other species of tick were collected from captures bank voles, they choose to focus solely on I. ricinus larvae, though it is not clear why this decision was made nor was it stated what other species of tick were found on the captures bank voles. Additionally, while environmental conditions were mentioned as serving as a defining factor in tick activity and therefore burden size, I am left wondering what was so different about 2018 that it impacted the burden size the way it did and it is my opinion that it would be to the benefit of the reader for the authors to include their judgement on why this occurred.

It is my opinion that this study serves dual importance. While not only ethically important to reduce the suffering/stress of a captured animal, it is advantageous to strive for studying animals in the most natural conditions possible—natural conditions in this study being a live host for the tick to feed upon. Especially when feeding behavior of ticks on live versus dead hosts is not well studied, these findings provide some insight into what conditions may be factoring into their drop-off behavior. Overall, I believe that the study was well conducted, and that the paper is well written, yielding insights beneficial to future studies on tick burden size and small mammal capture.

Reviewer #3: PLOS ONE

Manuscript Number: PONE-D-20-16817

AUTHORS: De Pelsmaeker et al.

TITLE: Do bank voles (Myodes glareolus) trapped in live and lethal traps show differences in tick burden size?.

SUMMARY: There are several methods that have been used to estimate tick burden (i.e., number of ticks per host), however comparative studies to evaluate methods are rare. This manuscript compares the efficacy of two forms of trapping methods to estimate larval Ixodes ricinus tick burdens on the bank vole, an important rodent reservoir species of several tick-borne diseases in Europe. In general, the manuscript was well written, the methods used to test the comparison and analyze the results were appropriate. The conclusions are generally sound. This paper will assist other researchers in choosing the most suitable sampling method for addressing their specific research objectives.

The following critique is intended to improve the manuscript.

MAJOR CRITICISM:

I do not know if this was the authors' fault, but Tables embedded within the text were formatted poorly, resulting in misaligned numbers or entire columns of data being truncated, making review of the manuscript unnecessarily difficult. For example: Table 2. Data for Year 2018, Summer and Autumn are not present; the columns lie outside the margin of the page. Likewise, Table 3, Column 5. There are typed-over numbers, some which completely obscure the p-values. What are these extra numbers and where do they belong? If they are extension of the p-values, please consider abbreviating p-values to 3 decimal places.

SOME STATISTICAL QUESTIONS REMAIN:

Table 3 indicates a significant interaction effect between trap type and season. Figure 2 indicates larval Ix. ricinus are most abundant in the spring, indicating that springtime is the most important season to sample voles for larval ticks in Norway. Figure 2 also indicates that during the springtime, voles captured by lethal traps had higher mean burdens of larval ticks than voles captured alive. Indeed, standard error bars around the means did not overlap. Did authors test for statistical significance in mean tick burdens between trap types for springtime ONLY (i.e., perform a t-test on log-transformed counts or nonparametric Wilcoxon signed rank test on untransformed counts)? If not, this should be done. If so, please report the statistical outcome.

Were there any differences between trap types and tick infestation PREVALENCE (i.e., % of voles parasitized with larval ticks)? A brief sentence or two to state the results should be included.

MINOR COMMENTS Bulleted points are listed in order of importance.

• Authors would benefit from reading (and citing) the recent review article concerning methods for estimating tick burdens on host animals – see: Lydecker HW, et al. 2019. Counting ticks is complex: A review and comparison of methods. J. Med. Entomol. 56: 1527-1533.

• Page 10, Table 1. Please add another column to this table at the far right-hand side to indicate the total number of voles for row 4 (Lifjell), row 5 (Laerdal), and row 6 (Totals). Heading for this new column should be “Total”.

• Line 317 states “Burden size increased with temperature”. But Fig. 2 shows the highest burden sizes were in the Spring, not the Summer. Typically, ambient temperatures are LOWER in the Spring than in the Summer. Was this not the case in Norway during 2017 and 2018? Perhaps, authors should state somewhere in the paper, what were the average daily temperatures at each location during May 20th – 30th, July 20th – 30th, and September 20th – 30th. If necessary, re-assess the statement that “burden size increased with temperature”.

• Line 26. Abstract. The species name of the tick is misspelled.

• Line 210. I do not think that a p-value of 0.01 indicates “borderline significance”. Please re-assess.

• Page 11, Table 3. Please list the parameters in some meaningful way. Suggest listing the parameters in order of their significance – i.e., p-values.

• Line 203. Values in parenthesis – (4.8 and 1.2, respectively). Please indicate the units for these values – i.e., (4.7 and 1.2 ticks per vole, respectively).

• Line 205-206. Something is missing at the end of this sentence. Suggest …”compared to 2017”.

• Lines 284 – 287. This sentence references Pollack (2012) study on lizards, followed immediately with reference to current study on voles. Recommend splitting the sentence into two sentences – one about lizards, the second about voles.

• Line 230, Figure 4 caption. Should read … “Mean numbers of male and female voles captured per trap.”

• Line 234. Table 2 title. Please change the word “ticks” to “larval Ixodes ricinus ticks”.

6. PLOS authors have the option to publish the peer review history of their article (what does this mean?). If published, this will include your full peer review and any attached files.

Reviewer #1: No

Reviewer #2: **Yes: **Anna R. Pasternak

Reviewer #3: **Yes: **Jefferson A. Vaughan

---

## [Author Response · Author response to Decision Letter 0]

10 Aug 2020

Thank you for your time and efforts in providing a review to the initial manuscript. We welcome all criticism and opportunities to improve the revised manuscript. You will find a point-by-point response to each of the reviewers' comments in the attached "Response to Reviewers" document.

---

## [Editor Report · Decision Letter 1]

31 Aug 2020

Do bank voles (Myodes glareolus) trapped in live and lethal traps show differences in tick burden?

PONE-D-20-16817R1

Dear Dr. De Pelsmaeker,

We’re pleased to inform you that your manuscript has been judged scientifically suitable for publication and will be formally accepted for publication once it meets all outstanding technical requirements.

Kind regards,

Brian Stevenson, Ph.D.

Academic Editor

PLOS ONE
---

## [Editor Report · Acceptance letter]

4 Sep 2020

PONE-D-20-16817R1 

Do Bank Voles (*Myodes glareolus*) Trapped in Live and Lethal Traps Show Differences in Tick Burden? 

Dear Dr. De Pelsmaeker:

I'm pleased to inform you that your manuscript has been deemed suitable for publication in PLOS ONE. Congratulations! Your manuscript is now with our production department. 

Kind regards, 

on behalf of

Prof. Brian Stevenson 

Academic Editor

PLOS ONE